# Abscisic Acid Perception and Signaling in *Chenopodium quinoa*

**Gastón Alfredo Pizzio** [1,2]

1   Instituto de Biología Molecular y Celular de Plantas, Universidad Politécnica de Valencia (UPV-CSIC), CP46022 Valencia, Spain; gpizzio@gmail.com or gaspiz@ibmcp.upv.es
2   CIT-RIO NEGRO Sede Atlántica, Universidad Nacional de Río Negro (UNRN-CONICET), CP8500 Viedma, Río Negro, Argentina

**Abstract:** Food production and global economic stability are being threatened by climate change. The increment of drought episodes and the increase of soil salinization are major problems for agriculture worldwide. *Chenopodium quinoa* (quinoa), as a resilient crop, is capable of growth in harsh environments due to its versatility and adaptive capacity. Quinoa is classified as an extremophile crop, tolerant to salinity, drought and low temperature. Furthermore, quinoa is recognized as a pseudo-cereal with outstanding nutritional properties. The phytohormone ABA is a key regulator of physiological responses to salinity and drought, among others stressful conditions. In this article we want to revise recent discoveries regarding ABA perception and signaling in quinoa, and evaluate its implications on stress-tolerance breeding of this pseudocereal and other crops.

**Keywords:** plant hormone; crops; ABA signaling; PYL; SnRK2s

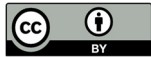

## 1. Introduction

In recent years, the production of quinoa (*Chenopodium quinoa Willd.*) has intensified due to the growing international demand [1]. The global interest in understanding, researching, producing and consuming this Andean grain was boosted as a result of the declaration of the International Year of Quinoa in 2013 by the United Nations Food and Agricultural Organization (FAO) [1,2]. Quinoa is recognized as a crop with exceptional nutritional properties [3,4]. Quinoa seeds contain high protein levels, and also provide all essential amino acids, in contrast to cereals [5]. Moreover, this pseudocereal is suitable for gluten-free diets, for instance for people with celiac disease [6]. Even more, cooked quinoa seeds have a low glycaemic index (GI), in the range from 35–53 [7]. Low GI diets and the concomitant insulin resistance reduction have been shown to reduce the risk of cardiovascular diseases and diabetes [7,8]. Furthermore, quinoa supplies elevated levels of vitamins, minerals, dietary fiber, and antioxidants [9,10]. In fact, the National Aeronautics and Space Administration (NASA) pointed out quinoa as a perfect food source for astronauts in long-term human space missions [11].

Importantly, as a resilient crop, quinoa is well adapted to grow in harsh environments. Traditionally cultivated in the highlands and inter-Andean valleys [12], quinoa production is spreading worldwide. Moreover, quinoa has enormous potential for expansion and production due to its versatility and adaptive capacity to grow in different environmental niches [13]. Quinoa shows high tolerance to salinity, drought, and low temperature [13,14]. Furthermore, as a facultative halophyte, quinoa growth performance is optimum under 100 mM NaCl conditions, and shows mild growth reduction under high salinity levels (400–500 mM NaCl, near sea-water level) [13,15]. Quinoa also has low water requirements and it is considered naturally tolerant to drought. The main physiological characteristics of quinoa regarding drought tolerance are plasticity in growth, low osmotic potential, low reduction in turgid weight/dry weight ratio, and maintenance of sustained positive turgor pressure at reduced water potentials [16,17]. WUE is high in quinoa, balancing its decreased leaf stomatal conductance, which leads to

carbon gain optimization. Photosynthesis is maintained even after abscisic acid (ABA) production and stomatal closure in response to mild water deficit [18]. Of note, quinoa has a high basal level of ABA response [19].

The ubiquitous phytohormone ABA regulates a myriad of physiological processes in vascular plants, such as seed and bud dormancy [20], organ development [21], and fruit ripening [22,23]. Further, ABA is a key regulator of physiological responses to stressful conditions, such as salinity and drought [24–27]. For instance, ABA triggers stomatal closure [28], regulates the hydraulic response to sustained drought conditions [29], controls the carbohydrate metabolism recovery after stress [30], and modulates the salt exclusion when plants are under salinity stress [31]. ABA is a sesquiterpenoid synthesized following the carotenoid pathway in plastids and cytosol [32,33]. ABA is produced mainly by the roots and transported to shoots via the transpiration flow [34–38]; however, drought-induced production of ABA also occurs in the leaf mesophyll cells [39,40]. Consequently, signaling begins with ABA perception in the target cells through pyrabactin-resistance1-like (PYL) receptors, and the formation of a tripartite complex between ABA, PYL receptors, and clade A protein phosphatase type 2Cs (PP2Cs) [41–44]. As a consequence, the inhibition of ABA-activated subclass III SNF1-related protein kinases 2 (SnRK2s) by clade A PP2Cs is relieved [45,46]. Then, activated SnRK2s modulate the activity of different effectors such as transmembrane channels and transcription factors [47]. In a second layer of regulation, the activity as well as the half-life of this ABA core signaling pathway is further regulated by accessory kinases, the circadian system, and the proteasome system pathway [47–51].

It was proposed that increased ABA synthesis and enhanced transport and signaling may be critical for quinoa as an extremophile plant [19]. In this review we want to update and discuss the implications of recent discoveries made with respect to quinoa regarding ABA perception and signaling. This theoretical basis will help in stress-resistance breeding of quinoa and other crops.

## 2. ABA Perception and Signaling in Quinoa

ABA is a key player in plant drought and salinity tolerance [47,52,53]. The PYL receptors are the first component of the ABA core signaling pathway. These proteins perceive the hormone and form a ternary complex between ABA–PYL–PP2CA, triggering the signaling cascade and response. The PYL family is composed of several members, for instance 12 PYLs in palm [54], 14 PYLs in tomato [55], 12 PYLs in rice [56], 23 PYLs in *N. benthamiana* [57], 29 PYLs in tobacco [58], and 38 PYLs in wheat [59]. Recently, 20 PYLs were identified in quinoa (Figure 1) [60]. The phylogenetic classification of these 20 CqPYLs grouped the receptors in the three classical clades of ABA receptors: PYL8 subgroup (subfamily I), PYL4 subgroup (subfamily II), and PYL1 subgroup (subfamily III) [53,57,61–63]. Furthermore, the expression profiles analysis of *CqPYLs* in 1-week-old seedlings showed that *CqPYL8c-d* and *CqPYL4a-b* were the most abundant ABA receptors, followed by *CqPYL8a-b, CqPYL5a-b, CqPYL8e-f,* and *CqPYL1a-b* [60]. Furthermore, *CqPYLs* displayed strong tissue specificity: for instance, *CqPYL8c-d* showed high expression in shoots and adult leaves, while *CqPyl8ab, CqPYL4a-b, and CqPYL1a-b* were predominantly expressed in roots. In Arabidopsis, lateral root growth is regulated by PYL8 in the presence of ABA [63]. By contrast, in quinoa, several members of the three sub-families of CqPYL have the potential to participate in root growth and development in response to the ABA.

Quinoa salinity and drought tolerance seem to be in detriment of the adaptation to high temperature environments [14,64-65]. These traits are connected to stomatal closure and ABA signaling [12,66]. Moreover, drought and heat stress often coincide in time in a given environment and may trigger conflicting responses. Plant tolerance to high temperatures requires leaf cooling through stomatal opening [66,67], the opposite to salinity and drought tolerance, where the plant requires stomatal closure to avoid transpiration. Deeper studies on ABA-CqPYLs affinity and dynamics could shed light on this issue,

unveiling molecular mechanisms for the reduction in heat tolerance at the expense of gaining in salt and drought tolerance in quinoa.

Upon ABA perception by PYL receptors, a ternary complex is formed between ligand, receptor, and the phosphatase co-receptors clade A PP2Cs (Figure 1). These phosphatases are key negative regulators of the ABA core signaling pathway involved in SnRK2 dephosphorylation inactivation. The ternary complex conformation inhibits clade A PP2Cs activity and allows SnRK2s activation. Unfortunately, a detailed study on the PP2Cs family in quinoa has not been reported. However, a fine description of SnRK2s family in quinoa was recently published [68]. SnRKs are highly conserved serine/threonine (Ser/Thr) protein kinases that participate in a myriad of physiological processes [69,70].,SnRKs family is classified in three subfamilies, based on the conservation of its kinase activity domains: SnRK1, SnRK2, and SnRK3 [71]. In particular, the plant-specific SnRK2 family plays a key role in plant adaptation to osmotic stress [72,73] in both ABA-dependent and ABA-independent signaling pathways. In *Arabidopsis thaliana* there exist 10 AtSnRK2 genes [74]. In quinoa, the identification of 13 SnRK2 family members was reported recently (Figure 1) [68]. Interestingly, despite the fact that quinoa is an allotetraploid, the SnRK2 subfamily did not show gene duplication in the genome as in respect to diploid species such as *Arabidopsis thaliana*, *Oryza sativa,* and *Populus trichocarpa* (with 10, 10, and 12 SnRK2 members, respectively) [75]. This fact is in agreement with the idea of paleopoliploids or ancient polyploids [76,77]. It was proposed that several previously considered diploid plant species, such as Arabidopsis and rice, could actually be ancestral polyploids that have undergone diploidization [78,79]. This evidence points to polyploidization as a common mechanism to drive plant evolution. The SnRK2s subfamily is also divided into three clades: subclass I, II, and III. Moreover, activation of the subclass III members SnRK2.2/3/6 in Arabidopsis is ABA-dependent and requires ABA-mediated inhibition of clade A PP2C [80]. In quinoa, six *CqSnRK2* genes belong to Group III and the expression is induced by drought stress. These six SnRK2s (CqSnRK2.1, CqSnRK2.3, and CqSnRK2.6–CqSnRK2.9) could be potentially involved in ABA core signaling. However, the phylogenetic analysis showed that CqSnRK2.6 and CqSnRK2.9 are relatively distant to AtSnRK2.2/3/6 [68]. More studies are needed to address the actual role of each CqSnRK2 in ABA signaling. In the meantime, with the help of protein modeling, a structural analysis showing the spatial conservation of key amino acid residues involved in the interaction with PP2Cs would shed some light on this issue. On the other hand, *subclass III SnRK2s* expression is strongly induced by ABA in Arabidopsis and rice [81,82]. Interestingly, *SnRK2s* genes from the three subclasses can be induced by ABA in quinoa, possibly pointing to a higher complexity of the ABA-dependent signal transduction pathway [68]. In quinoa, it was also found that the expression patterns of some partially duplicated *SnRK2* genes, such as the pair *CqSnRK2.3/CqSnRK2.4*, were different. This indicates a functional divergence of homoeologous gene pairs during quinoa evolution, probably at the gene-promoter region [68,83]. Further understanding of the CqSnRK2s dynamic will requires more studies.

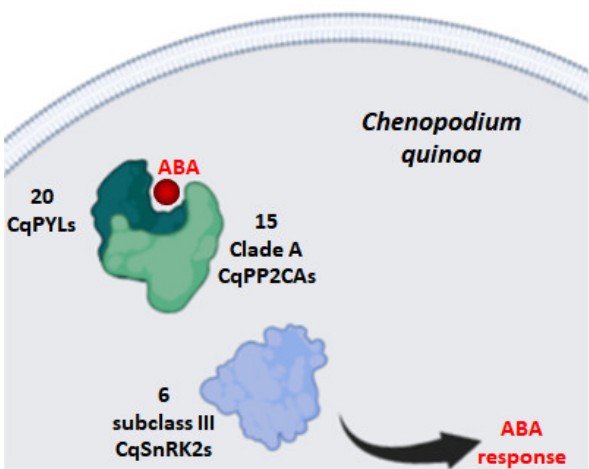

**Figure 1.** ABA perception and signaling in quinoa. ABA is perceived by the CqPYL receptors (20 in quinoa [60]). A ternary complex should be formed upon ABA perception between the phytohormone, CqPYLs, and CqPP2Cs (there could be approximately 15 ABA-related clade A PP2Cs as a rough estimation from annotations in quinoa genome.Finally, released subclass III CqSnRK2s must trigger ABA response (6 genes reported in [68]). Icons were obtained from Biorender.

### 3. Role of ABA in Quinoa's Bladder Cells

Salt stress causes major crop losses worldwide [84]. Soil salinization is affecting near one third of all irrigated lands [85], and the problem still growing, especially in arid and semi-arid regions. Halophytic plants are well adapted to growth, and complete their life cycle in such sub-optimal environments. Even more, some species, such as quinoa, show improved growth in the presence of NaCl [86,87]. The understanding of the driving mechanisms of salt tolerance in halophytes would be instrumental in crop improvement, facilitating future food production.

Epidermal bladder cells (EBCs) are modified leaf trichomes and represent a key morphological feature of quinoa, and other halophytes [88]. EBCs are pronounced on young leaves, and can contain up to 1 M of NaCl, serving as $Na^+$ dumps [89,90]. Moreover, mechanical removal (by brushing) of EBCs makes quinoa more sensitive to salinity [90]. Controversially, a recent work proposes that EBCs function as a $K^+$ reservoir, playing an important role in salt-stress tolerance independently of $Na^+$ accumulation [91].

ABA could play a key role maintaining EBCs function under high osmotic stress (Figure 2) [19,92]. Moreover, EBCs could also operate as an ABA source and export this phytohormone to other plant tissues to enhance systemic response to abiotic stresses [19]. Transcriptomic analysis revealed that the ABA synthesis pathway was upregulated in EBCs [19,92], in particular the transcript levels of *NCEDs* and *SDR* genes (encoding the rate-limiting enzymes for ABA biosynthesis). Furthermore, it was also showed that the expression levels of ABA transporters, such as *ABCG40* and *ABCG25*, are higher in EBCs in respect to other tissues. On the other hand, ABA perception, signaling, and response should be incremented in EBCs given that the majority of the *CqPYLs* ABA receptors are in high levels [19].

An ABA-dependent (at least partially) plant response to abiotic stress is the accumulation of compatible osmolytes. These osmoprotective compounds play an important role in several processes such as cytosolic metabolism protection from the toxic effect of NaCl, cell turgor maintenance by water uptake and retention, protection of the integrity of cellular structures, and prevention of enzyme denaturation [65,93]. Osmoprotectants in plants are grouped into three major categories: amino acids, polyol/sugars, and quaternary amines [90,94]. Proline is the major osmolyte accumulated under salt stress in quinoa EBCs [90]. Moreover, transcriptomic data revealed that EBCs in quinoa leaves could act as a proline factory and extrude this osmolyte by transport across the PM providing osmotic homeostasis to other cells [19]. The rate-limiting enzymes for proline

biosynthesis are 1-pyrroline-5-carboxylate synthetases (P5CSs), and in Arabidopsis and crops *P5CSs* transcription is activated by both osmotic stress and ABA [95–98].

On the other hand, salt and compatible osmolyte accumulation in EBCs is related to specific transporters, such as CqProT (for proline), CqHAK (for $K^+$), CqNHX (for $Na^+$), and CqCLC and CqNRT (for $Cl^-$) [19,99,100]. These transporters depend on $H^+$ cotransport to perform the ion or molecule translocation. This process is energized by the proton motive force (PMF) established by $H^+$-ATPases and $H^+$-PPases in both the plasma membrane (PM) and tonoplast to generate proton gradients across membranes to drive ion and osmolyte transport (Figure 2) [101–104]. In fact, the number of *PM-H⁺-ATPases* genes in the quinoa genome is 2–3-fold higher than the number of ortholog genes in other crops such *Beta vulgaris*, *Solanum lycopersicum*, *Spinacia oleracea L.*, *Solanum tuberosum*, or *Vitis vinifera* [19]. Furthermore, *PM-H⁺-ATPases* are highly expressed in the bladder cells with respect to other tissues [19]. Moreover, in the quinoa genome database there are three genes annotated as orthologs of the *Arabidopsis V-H⁺-PPase* (*AVP1- Arabidopsis Vacuolar Pyrophosphatase 1*) [105]. Probably, quinoa has evolved a robust ion homeostasis system to cope with high salinity levels.

In Arabidopsis, ABA induces stomatal closure through guard cell ion fluxes regulation [106,107]. For instance, SnRK2.6 activates the malate transporter anion channel QUAC1/ALMT12 [108,109], and also induces the activation of ion efflux transporters, such as SLAC1 and KUP6, for $Cl^-$ and $K^+$, respectively [107,110–112]. On the contrary, SnRK2.6 inhibits the $K^+$ influx transporter KAT1 through phosphorylation, which also affects stomatal movement [113]. Unfortunately, the role of ABA in the regulation of ion and osmolyte transporter activity in quinoa EBCs remains elusive.

Regarding PMF generation, in-silico evidence suggests that AVP1, and the concomitant vacuoles energization, can be regulated by the ABA signaling pathway, specifically by HAB1 [114]. However, a deep characterization on this topic has not yet been made. In this line of study, ABA is also capable of regulating PM-H⁺-ATPases [102]. For instance, ABA-activated BAK1 phosphorylates the PM-H⁺-ATPase AHA2, inducing cytoplasmic alkalinization, ROS production, $Ca^{2+}$ influx, and stomatal closure [115]. On the other hand, in the absence of ABA, PP2CA inhibits AHA2, dephosphorylating its key Thr947 and leading to the $H^+$ extrusion suppression [102]. ABA signaling activation, in response to osmotic stress or low ABA concentrations (0.1 μM), relieves AHA2 inhibition, then Thr947 is phosphorylated and $H^+$ extrusion is restored [116]. Moreover, the multiple phosphatase mutant Qabi2-2 showed a correlation between high apoplasmic $H^+$ extrusion and enhanced root growth and hydrotropic bending response [63,116]. These phenotypes agree with the idea that, in absence of phosphatases, the maintenance of AHA2-Thr947 in the phosphorylated-active state leads to the apoplasmic energization by permanent $H^+$ efflux, and cell-wall loosening and extension [116,117]. Future studies may investigate the cross-talk between ABA signaling and AHAs in order to understand EBCs function.

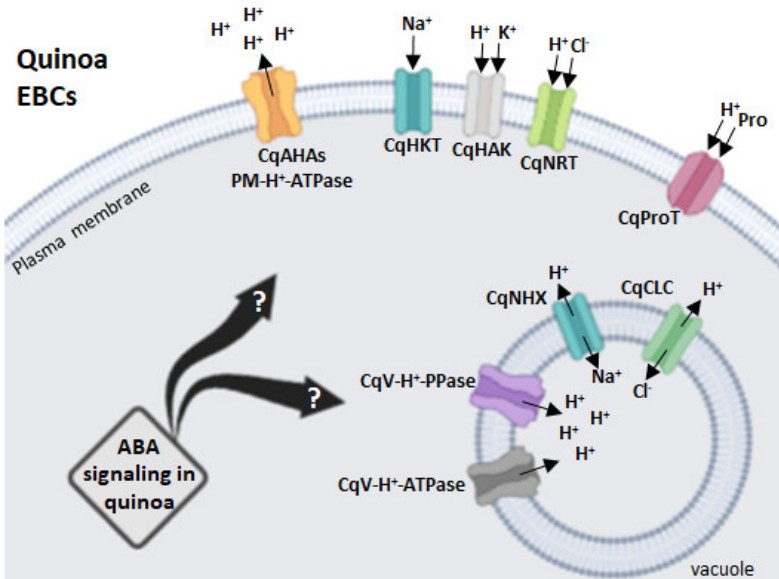

**Figure 2.** Potential transmembrane targets of ABA signaling in quinoa EBCs. CqHKT channels mediate Na$^+$ transport into EBCs. CqNHXs and CqCLC operate as antiporters of Na$^+$/H$^+$ and Cl$^-$/H$^+$, respectively. Potassium, chloride, and compatible osmolytes, such as proline (Pro), are transported into the sink cells by H$^+$ symporters (CqHAK, CqNRT and CqProT, respectively). Plasma membrane H$^+$ ATPase (CqAHAs/PM-H$^+$-ATPase), vacuolar H$^+$ ATPase (CqV-ATPase), and vacuolar CqV-H$^+$-PPase generate the proton motive force required for ions and solutes transport from apoplast to bladder cells cytoplasm, and then to the tonoplast. Icons were obtained from Biorender.

## 4. Conclusions and Future Perspectives

Increasing drought occurrences, freshwater resources declining, and soil salinization increments affect agriculture worldwide, including global economic stability and food security. These problems will grow because of global climate change. Quinoa is an exceptional crop, with the capacity to provide food security, especially in arid and semi-arid regions. Knowing more about the molecular mechanisms that sustain abiotic stress tolerance in quinoa will help in solving the above-mentioned problems. ABA perception and signaling represent a key molecular mechanism that regulates stress tolerance in plants. Some ABA-regulated physiological responses are similar under drought and salinity stress, such as stomatal closure, plant growth limitation, senescence induction, accumulation of compatible solutes, and protection against ROS [13,65,93]. The availability of quinoa genome assembly facilitates the beginning of ABA signaling characterization, among other processes [19,99,105]. Despite the fact that recently it was identified both the CqPYL and the CqSnRK2 families [60,68], the role of ABA in quinoa physiology is starting to emerge. Quinoa production demands an intensive domestication program such as the development of lower-height plants, more compact seed heads, and augmented heat and biotic stress tolerance [13]. To know more about the role of ABA in quinoa physiology will be useful in this regard. On the other hand, future investigations must be focused to determine the role of ABA in EBCs development and function, given that these are key cells related to salt and drought tolerance. Furthermore, ABA levels in EBCs are high, and the nutraceutical use of this natural source of phytohormone could be studied, since ABA also plays a role in the mammalian central nervous system as a neurotrophic factor, ameliorating some disorders related to sleep, depression, pain and memory [118].

**Author Contributions:** All authors have read and agreed to the published version of the manuscript.

**Funding:** This research received no external funding.

**Conflicts of Interest:** The authors declare no conflicts of interest.

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
