# Peer review of "Abscisic Acid Perception and Signaling in Chenopodium quinoa"

_stresses, doi:10.3390/stresses3010003_

Round 1

Reviewer 1 Report

The review manuscript Stresses-2096853 deals with very interesting and important topic of ABA signalling system in quinoa, plants species very tolerant to high salinity and drought. The manuscript is written in simple style with good scientific background and balanced references. Minor but important revision is required, mostly terminology and more clear statements in some cases. If the authors can address the points indicated below adequately and properly, the revised manuscript can be accepted for the publishing.

Minor comments and corrections:

(1) The main concern is for the last fragment in L149-173 about proton pumps, which has to be revised for more accurate information and statements. In general, author has to be very careful describing both enzymes (H+-ATPase and H+PPase) located either in plasma membrane or in tonoplast (vacuole). In the first case, for example, the proton pump enzyme has ti be designated as ‘PM H+-ATPase’, as it if perfectly present in the reference [102]. In contrast, in the second case, similar enzymes designated with letter ‘V’ (vacuolar), for example, ‘V-H+-PPase’. In L152, author indicated both ‘PM and tonoplast’ while in L155 only ‘PM’ was described. However, in all other cases, the location of these enzymes remains unclear. Therefore, during revision of this section, author has to insert either ‘PM’ or ‘V-‘ to the names of described enzymes. In case of both membranes, the description would be fine as it is present in L152. Please make corrections.

(2) In the same regard, Figure 2 has to be improved. Genes in plasma membrane have to be indicated as PM (but not ‘P-’), for example as ‘PM H+-ATPase’. I suppose that a big cycle in the right-bottom part of the Figure 2 is vacuole. If so, please indicate it in the Figure. Both genes for proton pump in vacuole have to be identified with prefix ‘V-’, as it was correctly indicated for ‘V-H+ATPase’ (but H+ was lost in the name). In contrast, the second gene in vacuole membrane, ‘V-H+-PPase’ has lost ‘V-‘. Please make corrections.

(3) L156-157. “Moreover, in quinoa there are 3 ortholog genes of the Arabidopsis H+-PPase (AVP1- Arabidopsis Vacuolar Proton-pump 1) [105].” This sentence contains several mistakes. Firstly, in the reference [105], there is no information (both in main text and in Supplementary material) about number of orthologous genes of H+-PPase in quinoa genome. Please check and provide more precise information and correct reference.

(4) L156-157. The second point is about full name of abbreviated ‘AVP1’. The full name in the brackets after AVP1 is incorrect and ‘proton pump’ is the name of a group of genes regarding their function and includes both H+-ATPase and H+-PPase. The correct name of AVP1 gene is ‘Arabidopsis vacuolar pyrophosphatase’ [Ferjani et al., 2011, Plant Cell 23, 2895-2908. DOI: 10.1105/tpc.111.085415]. This is not my paper and, therefore, this is not a ‘self-advertising’. After proven usage of a correct reference, this sentence has to be revised as follows: ‘Moreover, in quinoa there are X ortholog genes of the arabidopsis V-H+-PPase (AVP1, Arabidopsis vacuolar pyrophosphatase 1) playing important role as a proton pump into vacuole [YYY]’. Of course, author can choose another variant for the changes.

(5) L153 and L165. The term ‘AHA-H+-ATPases’ is used incorrectly. The abbreviation ‘AHA’ was introduced and explained in [102] as follows “arabidopsis PM H+-ATPase (AHA)”. Therefore, ‘AHA’ is already containing ‘H+-ATPase’ and it can be used either ‘AHA’ (or AHA2) or ‘arabidopsis PM H+-ATPase’ but not mix together.

Other minor notes and errors as follows:

(6) L13, L17, L39. The author has to be consistent and do not mix two terms, ‘tolerance’ and ‘resistance’. They are very similar but not synonyms for biological meaning. I strongly recommend the author change and use only ‘tolerance/tolerant’ (and ‘sensitivity/sensitive’) to abiotic stresses, like salinity and drought, as it was correctly used in L42 and L43. This is because ‘resistance/resistant’ (and ‘susceptibility/susceptible’) is typically used for disease resistance.

(7) L93. Please check that all botanical names, like Arabidopsis thaliana is written in Italics, like two lines below (L95). As well as all names of genes have also to be in Italics.

(8) L94. The following phrase is not perfect: “…did not show GENE AMPLIFICATION in the genome…”. I emphasize ‘gene amplification’ is upper case because I suppose the author wants to say ‘gene duplication’ in genome due to polyploidy process. Please check and correct.

(9) L132. Could you please clarify your statement with the reference [91] as follows: “…EBCs can also function as K+ reservoir, playing an important role in salt-stress tolerance…”. If this is true, what is about Na+ accumulation in EBC, because it can reach up to 1M NaCl for Na+ ions dumps, as described two lines above? Does this mean that EBC accumulates both Na+ and K+ in high concentrations? If so, this is very conflicting statement because sodium and potassium ions have antagonistic effects in plant cell during salt stress. Please clarify and add a sentence or two with your explanation in the text.

(10) L189. Please avoid ‘colloquial jargon’ like “smart decision” regarding quinoa. Please replace it for more scientific term or phrase.

(11) L195-196. The author’s statement about ‘ABA-related phenotypes’ as “…development of shorter plants with fewer branches and more compact seed heads…” needs more proof and clarification. The included reference [13] is a book, while citations for a chapter or chapters in this book or other papers with more precise information about ‘ABA-related phenotypes’ in quinoa is required.

Author Response

Minor comments and corrections:

(1) The main concern is for the last fragment in L149-173 about proton pumps, which has to be revised for more accurate information and statements. In general, author has to be very careful describing both enzymes (H+-ATPase and H+PPase) located either in plasma membrane or in tonoplast (vacuole). In the first case, for example, the proton pump enzyme has ti be designated as ‘PM H+-ATPase’, as it if perfectly present in the reference [102]. In contrast, in the second case, similar enzymes designated with letter ‘V’ (vacuolar), for example, ‘V-H+-PPase’. In L152, author indicated both ‘PM and tonoplast’ while in L155 only ‘PM’ was described. However, in all other cases, the location of these enzymes remains unclear. Therefore, during revision of this section, author has to insert either ‘PM’ or ‘V-‘ to the names of described enzymes. In case of both membranes, the description would be fine as it is present in L152. Please make corrections.

Corrections were made accordingly.

(2) In the same regard, Figure 2 has to be improved. Genes in plasma membrane have to be indicated as PM (but not ‘P-’), for example as ‘PM H+-ATPase’. I suppose that a big cycle in the right-bottom part of the Figure 2 is vacuole. If so, please indicate it in the Figure. Both genes for proton pump in vacuole have to be identified with prefix ‘V-’, as it was correctly indicated for ‘V-H+ATPase’ (but H+ was lost in the name). In contrast, the second gene in vacuole membrane, ‘V-H+-PPase’ has lost ‘V-‘. Please make corrections.

Corrections were made accordingly.

(3) L156-157. “Moreover, in quinoa there are 3 ortholog genes of the Arabidopsis H+-PPase (AVP1- Arabidopsis Vacuolar Proton-pump 1) [105].” This sentence contains several mistakes. Firstly, in the reference [105], there is no information (both in main text and in Supplementary material) about number of orthologous genes of H+-PPase in quinoa genome. Please check and provide more precise information and correct reference.

Corrections were made accordingly.

The three AVP1 orthologous genes are annotated in the quinoa genome database generated in [105]. In fact there are five genes annotated but two of them are too short to give a functional peptide (see the information of these genes in the table shown below).

(4) L156-157. The second point is about full name of abbreviated ‘AVP1’. The full name in the brackets after AVP1 is incorrect and ‘proton pump’ is the name of a group of genes regarding their function and includes both H+-ATPase and H+-PPase. The correct name of AVP1 gene is ‘Arabidopsis vacuolar pyrophosphatase’ [Ferjani et al., 2011, Plant Cell 23, 2895-2908. DOI: 10.1105/tpc.111.085415]. This is not my paper and, therefore, this is not a ‘self-advertising’. After proven usage of a correct reference, this sentence has to be revised as follows: ‘Moreover, in quinoa there are X ortholog genes of the arabidopsis V-H+-PPase (AVP1, Arabidopsis vacuolar pyrophosphatase 1) playing important role as a proton pump into vacuole [YYY]’. Of course, author can choose another variant for the changes.

Corrections were made accordingly.

(5) L153 and L165. The term ‘AHA-H+-ATPases’ is used incorrectly. The abbreviation ‘AHA’ was introduced and explained in [102] as follows “arabidopsis PM H+-ATPase (AHA)”. Therefore, ‘AHA’ is already containing ‘H+-ATPase’ and it can be used either ‘AHA’ (or AHA2) or ‘arabidopsis PM H+-ATPase’ but not mix together.

Corrections were made accordingly.

Other minor notes and errors as follows:

(6) L13, L17, L39. The author has to be consistent and do not mix two terms, ‘tolerance’ and ‘resistance’. They are very similar but not synonyms for biological meaning. I strongly recommend the author change and use only ‘tolerance/tolerant’ (and ‘sensitivity/sensitive’) to abiotic stresses, like salinity and drought, as it was correctly used in L42 and L43. This is because ‘resistance/resistant’ (and ‘susceptibility/susceptible’) is typically used for disease resistance.

Corrections were made accordingly.

(7) L93. Please check that all botanical names, like Arabidopsis thaliana is written in Italics, like two lines below (L95). As well as all names of genes have also to be in Italics.

Corrections were made accordingly.

(8) L94. The following phrase is not perfect: “…did not show GENE AMPLIFICATION in the genome…”. I emphasize ‘gene amplification’ is upper case because I suppose the author wants to say ‘gene duplication’ in genome due to polyploidy process. Please check and correct.

Corrections were made accordingly.

(9) L132. Could you please clarify your statement with the reference [91] as follows: “…EBCs can also function as K+ reservoir, playing an important role in salt-stress tolerance…”. If this is true, what is about Na+ accumulation in EBC, because it can reach up to 1M NaCl for Na+ ions dumps, as described two lines above? Does this mean that EBC accumulates both Na+ and K+ in high concentrations? If so, this is very conflicting statement because sodium and potassium ions have antagonistic effects in plant cell during salt stress. Please clarify and add a sentence or two with your explanation in the text.

Clarifications were made accordingly.

 (10) L189. Please avoid ‘colloquial jargon’ like “smart decision” regarding quinoa. Please replace it for more scientific term or phrase.

Corrections were made accordingly.

(11) L195-196. The author’s statement about ‘ABA-related phenotypes’ as “…development of shorter plants with fewer branches and more compact seed heads…” needs more proof and clarification. The included reference [13] is a book, while citations for a chapter or chapters in this book or other papers with more precise information about ‘ABA-related phenotypes’ in quinoa is required.

Corrections were made accordingly.

Reviewer 2 Report

Dear Authors

The review is interesting and deserves consideration in this journal. The content is correctly organized and clear. I suggest the author to consider more biological data.

Author Response

Unfortunately, biological data regarding ABA perception and signaling in quinoa scarse. This field of research particularly in quinoa is starting to arise. I hope this review helps to trigger new investigations in this regard.